# Monte Carlo Models for Sub-Chronic Repeated-Dose Toxicity: Systemic and Organ-Specific Toxicity

**DOI:** 10.3390/ijms23126615

**Published:** 2022-06-14

**Authors:** Gianluca Selvestrel, Giovanna J. Lavado, Alla P. Toropova, Andrey A. Toropov, Domenico Gadaleta, Marco Marzo, Diego Baderna, Emilio Benfenati

**Affiliations:** Laboratory of Chemistry and Environmental Toxicology, Department of Environmental Health Sciences, Istituto di Ricerche Farmacologiche Mario Negri IRCCS, Via Mario Negri 2, 20156 Milan, Italy; alla.toropova@marionegri.it (A.P.T.); andrey.toropov@marionegri.it (A.A.T.); domenico.gadaleta@marionegri.it (D.G.); marco.marzo@marionegri.it (M.M.); diego.baderna@alice.it (D.B.); emilio.benfenati@marionegri.it (E.B.)

**Keywords:** NOAEL, LOAEL, QSAR, sub-chronic repeated-dose toxicity, organ-specific toxicity

## Abstract

The risk-characterization of chemicals requires the determination of repeated-dose toxicity (RDT). This depends on two main outcomes: the no-observed-adverse-effect level (NOAEL) and the lowest-observed-adverse-effect level (LOAEL). These endpoints are fundamental requirements in several regulatory frameworks, such as the Registration, Evaluation, Authorization and Restriction of Chemicals (REACH) and the European Regulation of 1223/2009 on cosmetics. The RDT results for the safety evaluation of chemicals are undeniably important; however, the in vivo tests are time-consuming and very expensive. The in silico models can provide useful input to investigate sub-chronic RDT. Considering the complexity of these endpoints, involving variable experimental designs, this non-testing approach is challenging and attractive. Here, we built eight in silico models for the NOAEL and LOAEL predictions, focusing on systemic and organ-specific toxicity, looking into the effects on the liver, kidney and brain. Starting with the NOAEL and LOAEL data for oral sub-chronic toxicity in rats, retrieved from public databases, we developed and validated eight quantitative structure-activity relationship (QSAR) models based on the optimal descriptors calculated by the Monte Carlo method, using the CORAL software. The results obtained with these models represent a good achievement, to exploit them in a safety assessment, considering the importance of organ-related toxicity.

## 1. Introduction

The European Commission (EC) requires continuous efforts to develop new approaches in the safety assessment of chemicals consistent with the regulatory framework. The European Directive on cosmetics, for example, laid out a new vision in Europe for the safety assessment of cosmetics, moving from the consolidated, animal-based, toxicological approach to a novel paradigm that bans the use of animals for toxicity tests [1,2]. Regulators explicitly require a deep examination of RDT, since it gives crucial details on the critical adverse effects resulting from repeated exposure to a certain substance in a limited period [3,4]. REACH, which prescribes within its annexes an assessment on RDT [3], firmly supports the application of new-approach methodologies in order to provide forward-looking solutions in the field of risk assessment.

RDT studies usually employ a wide array of standards; indeed, they present different exposure times and routes as well as various models, such as rodent or non-rodent species [5]. The exposure times can be divided into three groups: (i) sub-acute toxicity studies, which have a duration of up to 28 days (usually 2–4 weeks); (ii) sub-chronic toxicity, which has a duration of about 90 days; (iii) chronic (long-term) toxicity (repeated doses over a major part of the life-span) [4].

As toxicologically relevant pieces of evidence, the NOAEL and LOAEL are derived from the RDT studies. The NOAEL is the highest experimental dose at which no significant adverse response is observed, and the LOAEL is the lowest dose at which adverse effects arise compared to a control group [6]. NOAEL can be used to determine the important toxicological values considered in the evaluation of human exposure, such as the acceptable daily intake (ADI) and the reference dose (RfD) [7].

In the safety evaluation of cosmetics, when no benchmark dose (BMD) can be calculated, NOAEL is used to calculate the margin of safety (MoS) [4]—an essential parameter in the product information file (PIF)—which is vital for placing the product in the European market [2] and fundamental for indicating the potential toxicity for human health of substances in a cosmetic formulation [4].

NOAEL and LOAEL are marked by a substantial uncertainty since they are not intended for the dose−response curves but as single-dose estimates that largely depend on the experimental design [6]. As a result, other approaches such as the BMD have been proposed in place of the classical methodologies [4,8]. Unfortunately, the availability of data useful for BMDs is quite limited.

RDT studies require high costs and considerable time expenditures. For this reason, in silico methods are needed to offer a valid, future-oriented alternative [9]. Moreover, only a few attempts have been made to apply QSAR methods to simultaneously model NOAEL and LOAEL [6,10,11].

The aim of this work was to develop new QSAR models to predict NOAEL and LOAEL, focusing in particular on organ-specific toxicity, to obtain fundamental information on systemic toxicity and the advanced solutions in the research used for the in silico models. To achieve this, we used the Monte Carlo algorithm, with the CORAL software (http://www.insilico.eu/coral/, accessed on 11 June 2021). Other attempts to develop models for NOAEL and LOAEL using CORAL have been made in recent years [6,10,11]. In this work, we extended the method by applying the algorithm to new data related to organs, particularly the liver, kidney and brain, focusing on the toxicity of a specific organ. The models will be implemented on the VEGA platform (https://www.vegahub.eu, accessed on 2 December 2021) and will be freely available.

Finally, these models will be used to refine the battery of models within the VERMEER Cosmolife software [2] (https://www.life-vermeer.eu/download-software/, accessed on 3 May 2022), a new tool developed in the LIFE VERMEER project (LIFE16 ENV/IT/00016—https://www.life-vermeer.eu/, accessed on 3 May 2022). In particular, the section of the tool related to the dose–response assessment will be improved, to boost the potential of the tool and provide new solutions in the field of risk assessment [2].

## 2. Results

### 2.1. Characterization of the Datasets

The datasets contain chemicals from different sources covering several functional categories, such as pesticides, drugs, food additives and cosmetics. Figure 1 shows the histograms of the ToxPrint chemotypes identified in the datasets, reported in Appendix A. The chemotypes matching the structures in the datasets are shown if the frequency is over 10%. Out of 729 chemotypes, respectively, 376, 318, 211 and 328 were found in the general, kidney, brain and liver datasets. The most frequent ToxPrint chemotypes in all datasets were aromatic benzenes, C=O carbonyl generic, halides, alkanes, carboxamide, aminocarbonyl, heterocycles, aliphatic–aromatic ether and aromatic amines, all with over a 16% frequency in the datasets. The chemical space, covered by chemicals in terms of the structural and physico-chemical properties, is described in Appendix A. Datasets present a similar structural and physico-chemical profile. This represents important evidence, which allows us to say that these models could be suitable for predicting external data with a chemical space in line with what we defined here.

### 2.2. Models Developed

The CORAL models were based on the hybrid optimal descriptors extracted from the simplified molecular-input line-entry system (SMILES) and molecular graph. The models for the predictions of NOAEL and LOAEL in rats for sub-chronic repeated-dose toxicity are given in Equations (1)–(8), while Table 1 shows the statistical characteristics of the models. We calculated the validation metrics for each model, such as the determination coefficient (R^2^), the leave-one-out (LOO) cross-validated determination coefficient (Q^2^), predictive R^2^ (R^2^ pred), the criteria of predictability Q^2^_F3_, the concordance correlation coefficient (CCC), the index of ideality of correlation (IIC), the mean absolute error (MAE), the root mean squared error (RMSE) and the Fischer F-ratio (F) [12,13,14]. The predictions and experimental values, together with the extracted attributes of the models, are reported in Appendix A.

The most important criterion for assessing the predictive potential of a model is the statistical quality of an external validation set. In the general models, the predictive ability R^2^ in the validation sets reached 0.55 and 0.53, respectively, for NOAEL and LOAEL. The external validation metrics of the CCC and Q^2^_F3_ for NOAEL were 0.69 and 0.53, and 0.63 and 0.32 for LOAEL. The LOO cross-validation technique Q^2^ was used to assess the robustness of the models: it ranged from 0.50 to 0.54 for NOAEL and 0.49 to 0.54 for LOAEL. A value of 0.5 is an acceptable threshold for the Q^2^ reported in the literature [13].

The performances were better in the models for the organ-specific toxic effects than in the general models. In the kidney models, the predictive ability R^2^ in the validation sets reached 0.59 for NOAEL and 0.69 for LOAEL. The external validation metrics, CCC and Q^2^_F3_, were 0.75 and 0.84 for NOAEL, and 0.82 and 0.81 for LOAEL. The Q^2^ ranged from 0.48 to 0.58 for NOAEL and from 0.49 to 0.63 for LOAEL. In the brain models, R^2^ in the validation sets reached 0.53 for NOAEL and 0.69 for LOAEL. The agreement between the experimental and calculated values, expressed by the metric CCC, was 0.67 for NOAEL and 0.80 for LOAEL. The criterion of predictability Q^2^_F3_ was 0.81 for NOAEL and 0.82 for LOAEL. Moreover, Q^2^ ranged between 0.46 and 0.51 for NOAEL and 0.44 and 0.61 for LOAEL. Last but not least, in the liver models, R^2^ reached 0.55 and 0.61 in the validation sets for NOAEL and LOAEL, respectively. The metric CCC reached similar values for NOAEL and LOAEL. The criterion of predictability Q^2^_F3_ was 0.55 for both NOAEL and LOAEL. The robustness of the models expressed by Q^2^ reached similar values: higher than 0.72 for NOAEL and LOAEL. It can be noticed that LOAEL performs better than NOAEL in all the cases. Considering the complexity of modeling these endpoints, the overall performance is quite satisfactory for all the models. It is possible to observe that the performance of the liver models is better than the other organ-specific models, showing higher robustness, expressed by Q^2^, and lower MAE and RMSE values.

#### 2.2.1. General Models

The general NOAEL model was built up using the equivalent distribution in the active training set (n = 140), passive training set (n = 140), calibration set (n = 140), and validation set (n = 141). The model was calculated as follows:(1)NOAEL =2.1123367 (± 0.0064558)+0.1651535 (± 0.0010034)× DCW(1, 30)

The general LOAEL model was built up using the equivalent distribution in the active training set (n = 142), passive training set (n = 142), calibration set (n = 137), and validation set (n = 137). The model was calculated as follows:(2)NOAEL =2.1123367 (± 0.0064558)+0.1651535 (± 0.0010034)× DCW(1, 30)

#### 2.2.2. Kidney Models

The NOAEL kidney model was built up using the equivalent distribution in the active training set (n = 95), passive training set (n = 95), calibration set (n = 45), and validation set (n = 45). The model was calculated as follows:(3)NOAEL−kidney =2.1060212 (± 0.0059588)+0.1735682 (± 0.0016214)×DCW(1, 10)

The LOAEL kidney model was built up using the equivalent distribution in the active training set (n = 97), passive training set (n = 102), calibration set (n = 46), and validation set (n = 38). The model was calculated as follows:(4)LOAEL−kidney =2.4150252 (± 0.0053276)+0.1027443 (± 0.0010802)×DCW(1, 10)

#### 2.2.3. Brain Models

The NOAEL brain model was built up using the equivalent distribution in the active training set (n = 23), passive training set (n = 22), calibration set (n = 23), and validation set (n = 22). The model was calculated as follows:(5)NOAEL−brain =2.3962455 (± 0.0337524)+0.2029147 (± 0.0106681)× DCW(1,33)

The LOAEL brain model was built up using the equivalent distribution in the active training set (n = 22), passive training set (n = 23), calibration set (n = 22), and validation set (n = 23). The model was calculated as follows:(6)LOAEL−brain=2.5305357 (± 0.0383210)+0.1089317 (± 0.0053590)×DCW(1,4)

#### 2.2.4. Liver Models

The NOAEL liver model was built up using the equivalent distribution in the active training set (n = 97), passive training set (n = 94), calibration set (n = 30), and validation set (n = 31). The model was calculated as follows:(7)NOAEL−liver =1.7245212 (± 0.0040067)+0.0530130 (± 0.0002568)× DCW(2,15)

The LOAEL liver model was built up using the equivalent distribution in the active training set (n = 96), passive training set (n = 95), calibration set (n = 30), and validation set (n = 31). The model was calculated as follows:(8)LOAEL−liver =1.7148691 (± 0.0038133)+0.0529304 (± 0.0002685)× DCW(2,14)

## 3. Discussion

In the present study, the in silico models were developed for predicting NOAEL and LOAEL for the sub-chronic toxicity data (90 days of exposure). The development of the in silico models for RDT represents a challenging task [9,15]. The uncertainty that distinguishes the process derives, in particular, from the experimental design [6]. Indeed, different designs can produce contrasting NOAEL and LOAEL for the same chemical [9]. This variability makes the development of computational models extremely challenging. In recent years, only a few NOAEL and LOAEL models have been developed [6,10,15]. Table 2 shows the statistical characteristics that are in common with the models developed here, which we will discuss in Section 3.1. The complexity of developing models for NOAEL and LOAEL arises also from the fact that NOAEL and LOAEL are related to an extended range of effects that could arise in different target organs. These aspects, as well as the extensive modes and mechanism of actions, are difficult to consider separately, and often only partial information is provided. In this view, all of these features contribute to the final toxicity value. This study proposed new models for examining the critical effects that could arise in target organs, bearing in mind the abovementioned criticisms and limitations. To the best of our knowledge, this is one of the few attempts to develop NOAEL and LOAEL models for organ-specific toxicity. Predicting target-organ toxicity is a focal point in the next-generation risk assessment (NGRA) [16] and in the future, predictions of organ-level effects, in particular for the liver, should represent the basis of in silico modeling [17]. Moreover, data with a more defined understanding of the mechanisms/effects and/or adverse outcome pathways (AOP) is needed to improve the prediction strategy [17]. A thorough investigation of the ADME processes of a chemical, i.e., its ability to penetrate certain biological barriers and its distribution within the human body, is fundamental in the forward-looking NGRA approach [16]. Additionally, the physiologically-based kinetic (PBK) models, which are fundamental in determining the exposure to possible toxicants, can provide fundamental indications of concerns linked to target organs [18]. Considering this context, the QSAR models for organ-specific toxicity, which we have developed in this work, can provide outcomes useful for creating a future integrated-testing strategy.

Finally, our work follows the regulatory updates and aims to improve the risk-assessment procedure, introducing new alternative methods. NOAEL and LOAEL are fundamental endpoints for human health risk assessment under different regulatory contexts, such as cosmetics and REACH [3,4]. A significant use of time and money characterizes the in vivo RDT studies as well as the substantial number of animals involved. According to the cosmetics regulation, the in vivo RDT studies are not part of the testing procedures anymore [1], and REACH [3] calls for research to reduce the number of animals, according to the 3R principle [19]. Therefore, there is a pressing demand for new and valid alternatives for these endpoints.

However, the uncertainty of the NOAEL and LOAEL values, which is strongly dependent on the experimental design, tends to be reflected in the predictions obtained with the in silico models. Moreover, the link between the descriptors and systemic effects could not be linearly explained, since there is no weighty mechanistic principle [20]. Despite all these limitations, a reliable predictivity for NOAEL and LOAEL may be obtained using a combination of several techniques, such as QSARs, read-across, physiologically-based pharmacokinetic modeling (PBPK), the threshold of toxicological concern (TTC), and in vitro methods.

### 3.1. Comparison with Other Models

To the best of our knowledge, there are no NOAEL and LOAEL regression models related to organ-specific toxicity reported in the literature. We will therefore compare the general NOAEL and LOAEL models with the published and implemented models in free software.

In the last few years, other models have been developed for NOAEL and LOAEL in rats for sub-chronic repeated-dose toxicity (90 days of exposure). One model is currently freely available for predictions in the VEGA software [10,21]. The authors in [10] looked to studies in rodents to build NOAEL models. They considered studies with values for 28 days of treatment, dividing by 3 in order to approximate 90 days [4]. The dataset collected consisted of 140 chemicals with toxicity data from the Integrated Risk Information System (IRIS), Hazard Evaluation Support System (HESS) and Munro databases (https://www.epa.gov/iris, accessed on 5 January 2021) and from the OECD QSAR Toolbox, version 3.2 (https://qsartoolbox.org/, accessed on 5 January 2021). The performance of the model [10] is reported in Table 2.

In 2020, two new models (NOAEL and LOAEL) were developed [6]. These models were built using toxicity data from the Fraunhofer RepDose database, a non-public database. Studies only for the sub-chronic data from 90 days of oral toxicity in rats (Rattus norvegicus) were included. The models were based on 326 compounds. The statistical characteristics of the base model and the model based on chemicals inside the applicability domain (AD) are reported in Table 2. In this work, the models are only based on the public sources collected recently, covering more compounds than the other models reported in Table 2. The presence of the various functional groups and the chemical domains described by the physico-chemical properties, highlight the chemical heterogeneity of our datasets.

The most significant criterion for comparing the predictive potential of the models considers the statistical quality of an external (validation) set. For the NOAEL model, the validation set in [6] consisted of 38 compounds reaching R^2^ = 0.65. When excluding the compounds outside the AD, the predictive capacity reached R^2^ = 0.69, with 33 compounds covered. In the present work, the NOAEL model was based on a validation set of 141 compounds and reached R^2^ 0.55. Excluding compounds outside the AD improved the R^2^, reaching 0.61 with a 77% coverage of compounds. The NOAEL model in [10] reached an R^2^ of, respectively, 0.60 and 0.61 in the whole validation set and in the validation inside the AD. However, the validation set had fewer compounds than the previous ones.

The LOAEL model in [6] reached R^2^ = 0.59. The predictive ability reached R^2^ = 0.62, excluding compounds outside the AD, with 33 compounds inside it. In this work, the LOAEL model was based on a validation set of 137 compounds and reached R^2^ 0.53. R^2^ was similar, excluding compounds outside the AD with 74% of compounds.

## 4. Materials and Methods

### 4.1. Dataset Collection and Preparation

The datasets used to develop the QSAR models were built using the NOAEL and LOAEL experimental sub-chronic (90 days of exposure) oral-toxicity data in rats, from different publicly available databases, in line with their conformity to the OECD guideline, 408 [22]. These sources include the Munro and HESS databases, both of which are available from the OECD QSAR Toolbox, version 4.4 (https://qsartoolbox.org/, accessed on 5 January 2021), the IRIS database from the U.S. Environmental Protection Agency (EPA) (https://www.epa.gov/iris, accessed on 5 January 2021), the COSMOS database [23] (https://www.ng.cosmosdb.eu/about, accessed on 5 January 2021), the U.S. EPA’s ToxRefDB, v. 2.0 [24], and the European Food Safety Authority (EFSA) OpenFoodTox database [25,26]. In order to ensure consistent data, we took into account only data from the sub-chronic toxicity studies of oral exposure in rats (90 days, but also 91 and 92 days in order to extend the number of data), removing variability and inconsistencies due to interspecies’ differences [27]. Further, 2D SMILES chemical structures were retrieved for each substance. In some cases, SMILES were already present in the databases, otherwise, they were retrieved using a semi-automated data curation and quality checking workflow, implemented in the KNIME Analytics Platform, v. 4 [28,29].

Once a full dataset was set up under these rules, we rejected inorganic compounds, metal complexes and data related to mixtures or polymeric structures. Ionized structures were neutralized and counterions were excluded. Duplicates were analyzed and detected by checking SMILES at the 2D level, and by the CAS number. For compounds with multiple data, the lowest value was preserved, adopting a conservative approach. The experimental values (in mg/kg body weight (bw) per day) were converted to a logarithmic scale. Finally, to make the data more consistent, only chemicals with an NOAEL equal to or lower than LOAEL were maintained.

Another step was to retrieve data related not only to general-systemic toxicity but also to organ-specific toxicity. Therefore, following the same procedure, we built one dataset for general NOAEL and LOAEL and separate datasets using NOAEL and LOAEL data where the indication of the adverse effect is associated with a specific target/organ for the liver, kidney and brain. This provides a general dataset of 573 unique chemicals. For the organ-related data, we obtained 353 chemicals for the liver, 289 for the kidney and 91 for the brain. The datasets containing IDs, CAS numbers, SMILES and experimental values are reported in the Appendix A. A further curation aimed at the development of a specific model will be described in the next sections. The complete structure of these datasets is shown in Table 3.

### 4.2. Characterization of the Datasets

The chemical structures of the molecules (Appendix A) were analyzed with the ChemoTyper tool, v. 1.0, which searches and identifies 729 ToxPrint chemotypes (https://www.chemotyper.org, accessed on 16 April 2021). The complete lists and histograms of the chemotypes identified in the datasets can be found, respectively, in Appendix A.

Chemicals were also analyzed on the basis of their physico-chemical properties. These were calculated using the ‘Molecular Properties’ node of the analytics platform, KNIME (https://www.knime.com/knime-analytics-platform, accessed on 16 April 2021). The properties included the molecular weight, hydrogen bond acceptors, hydrogen bond donors, rotatable bonds count, Lipinski’s rule of five, topological polar surface area, atomic polarizabilities, bond polarizabilities and MannholdlogP. Appendix A show the mean, maximum and minimum values of the selected properties for each dataset.

### 4.3. Further Curation

To build the models, we removed the compounds with a low-frequency distribution of NOAEL and LOAEL values, as they were considered noise. For the liver dataset, an analysis was done to identify the possible structural or response outliers and activity cliffs, employing the Small Dataset Curator tool, v. 1.0.0 [30,31] and the istSimilarity tool, v. 1.0.7 [32]. The final number of compounds and the range of the NOAEL and LOAEL endpoints are summarized in Table 3. Appendix A show the frequency distribution of the experimental values.

### 4.4. Model Development and Optimal Descriptors

We used the CORAL software, v. 2020 (www.insilico.eu/coral, accessed on 6 June 2021), to build the QSAR models with the Monte Carlo method. The models were based on the so-called hybrid optimal descriptor, accounting for the correlation weights (CW) of both the SMILES and hydrogen-suppressed molecular graph (HSG) attributes [33,34].

The datasets were split randomly into the active training, passive training, calibration and validation sets [33,35]. Each subset had its own task. The task for the active training set was to calculate and identify which correlation weights gave the largest correlation between the experimental and predicted endpoint for the active training set. The task for the passive training set was to analyze if these correlation weights gave a reasonable correlation coefficient for similar compounds. The calibration set was used to detect overtraining. The validation set was used to evaluate the predictive potential of the model, using substances never used in the phase of model building.

The general form of a CORAL model can be described by the following one-variable Equation (9):(9)$$\mathrm{Endpoint}=C_{0}+C_{1}\times \;\mathrm{DCW}\left(T^{*},N^{*}\right)$$
where C_0_ and C_1_ are the intercept and the slope, DCW is the optimal descriptor, T* and N* are the best threshold and the best number of epochs of the Monte Carlo optimization. The best threshold is an integer to classify a molecular feature as active or rare. The building process never involves rare features. Both thresholds give the best statistical performance for the calibration set.

The optimal descriptor—a function of molecular features—that we applied, comprises the following components, as listed in Equation (10):(10)$$\mathrm{DCW}\left(T,N\right)={\mathrm{DCW}}_{\alpha }\left(T,N\right)+{\mathrm{DCW}}_{\beta }\left(T,N\right)+{\mathrm{DCW}}_{\gamma }\left(T,N\right)+{\mathrm{DCW}}_{\delta }\left(T,N\right)$$
where,
$${\mathrm{DCW}}_{\propto }\left(T,N\right)=x1\times \sum \mathrm{CW}\left(S_{k}\right)+x2\times \sum \mathrm{CW}\left({\mathrm{SS}}_{k}\right)+x3\times \sum \mathrm{CW}\left({\mathrm{SSS}}_{k}\right)$$
$${\mathrm{DCW}}_{\beta }\left(T,N\right)=\sum \mathrm{CW}\left(\mathrm{EC}1_{k}\right)+\sum \mathrm{CW}\left(\mathrm{EC}2_{k}\right)+\sum \mathrm{CW}\left(\mathrm{EC}3_{k}\right)+\sum \mathrm{CW}\left({\mathrm{NNC}}_{k}\right)$$
$${\mathrm{DCW}}_{\gamma }\left(T,N\right)=y1\times \overset{}{\underset{}{\sum }}\mathrm{CW}\left(\mathrm{APP}\left[\mathrm{Cl},N\right]\right)+y2\times \overset{}{\underset{}{\sum }}\mathrm{CW}\left(\mathrm{APP}\left[\mathrm{Cl},O\right]\right)+\;y3\times \overset{}{\underset{}{\sum }}\mathrm{CW}\left(\mathrm{APP}\left[\mathrm{Cl},S\right]\right)+y4\times \overset{}{\underset{}{\sum }}\mathrm{CW}\left(\mathrm{APP}\left[\mathrm{Cl},P\right]\right)$$
and
$${\mathrm{DCW}}_{\delta }\left(T,N\right)=\sum_{i}\sum_{j=i+1}M_{\mathrm{ij}}\;M=\begin{array}{cccccccccc} & F & \mathrm{Cl} & \mathrm{Br} & N & O & S & P & = & \#\\ F & & a_{12} & a_{13} & a_{14} & a_{15} & a_{16} & a_{17} & a_{18} & a_{19}\\ \mathrm{Cl} & & & a_{23} & a_{24} & a_{25} & a_{26} & a_{27} & a_{28} & a_{29}\\ \mathrm{Br} & & & & a_{34} & a_{35} & a_{36} & a_{37} & a_{38} & a_{39}\\ N & & & & & a_{45} & a_{46} & a_{47} & a_{48} & a_{49}\\ O & & & & & & a_{56} & a_{57} & a_{58} & a_{59}\\ S & & & & & & & a_{67} & a_{68} & a_{69}\\ P & & & & & & & & a_{78} & a_{79}\\ = & & & & & & & & & a_{89}\\ \# & & & & & & & & & \end{array}$$
where $a_{12}=\mathrm{CW}\left(\mathrm{APP}\left[F,\mathrm{Cl}\right]\right);{\;a}_{13}=\mathrm{CW}\left(\mathrm{APP}\left[F,\mathrm{Br}\right]\right);\ldots \;;{\;a}_{89}=\mathrm{CW}\left(\mathrm{APP}\left[=,\#\right]\right)$. Symbols ‘=’ and ‘#’ are indicators of the double and triple covalent bonds. Some examples of the atom-pairs proportions (APP) for APP[Cl,N] can be the following: (Cl.N)..1.1.A molecule contains one chlorine atom and one nitrogen atom;(Cl.N)..1.2.A molecule contains one chlorine atom and two nitrogen atoms;(Cl.N)..2.1.A molecule contains two chlorine atoms and one nitrogen atom.

The others APP attributes are defined in the same manner. Table 4 contains the types of attributes involved in the models with corresponding references.

The correlation weights were based on the index of ideality of correlation (IIC) and on the correlation intensity index (CII), both of which were calculated using the compounds in the calibration set, as described in the literature [14,41].

Table 5 shows details of the eight models developed. Screenshots of the methods M1–M8, utilized to build up the models, are shown in Appendix A.

### 4.5. Definition of the Applicability Domain

The applicability domain (AD) implemented in CORAL is based on a probabilistic measure ${\mathrm{defect}}_{\mathrm{SMILES}}$, which measures the quality of the molecular features extracted from the compound [33,36]. It is defined according to the distribution of the molecular features (S_k_) extracted from SMILES or HSG in the active training and calibration sets. The defect of S_k_ was calculated as:(11)$$d_{k}=\frac{\left|P(S_{k})-P^{\prime }(S_{k})\right|}{N\left(S_{k}\right)+N^{\prime }\left(S_{k}\right)}$$
where P(S_k_) and N(S_k_) are, respectively, the probability and frequency of S_k_ in the active training set. P′(S_k_) and N′(S_k_) are, respectively, the probability and frequency of S_k_ in the calibration set.

The defect of the SMILES was calculated as:(12)$${\mathrm{defect}}_{\mathrm{SMILES}}=\overset{\mathrm{NA}}{\underset{k=1}{\sum }}d_{k}$$
where NA is the number of non-blocked molecular features extracted from SMILES or HSG.

The criterion for the reliability of a prediction is defined as below in Equation (13):(13)$${\mathrm{defect}}_{\mathrm{SMILES}}<2\times \bar{D}$$
where $\bar{D}$ is the average of the statistical SMILES-defect for the compounds in the active training set [33]. Compounds having ${\mathrm{defect}}_{\mathrm{SMILES}}$ greater than the calculated threshold $2\times \bar{D}$ were considered outside the AD. In contrast to a traditional interpretation of the applicability domain, this approach provides a semi-qualitative criterion, named the statistical defect. Indeed, this is a measure of the presence of the molecular features observed in SMILES. If the majority of the molecular features are rare, then the SMILES have a large defect, while if the majority of the molecular features are represented many times in both the training set and in the calibration set, the defect is small. Appendix A report the statistical quality and the calculated threshold for the models applying the AD procedure described.

## 5. Conclusions

The present study provides new QSAR regression models for NOAEL and LOAEL, in particular, new models for NOAEL and LOAEL for the liver, kidney and brain. Considering the difficulty to obtain a robust model for these endpoints, predictions are quite satisfying, particularly for the organ-specific models. This is encouraging, with a view to a future integrated approach, where the prediction of NOAEL and LOAEL can be used to support a new multi-tier assessment strategy. Particularly convincing are the results for the liver, an organ of central importance for the risk assessment of various kinds of chemicals. These QSAR models will be implemented in the VEGA platform (https://www.vegahub.eu, accessed on 2 December 2021). The VEGA system will also show the six most similar substances and an evaluation of the reliability of the predictions. They have the important advantage of being freely available for the end users. Therefore, they can be used by the scientific community, worldwide, to support their activities. These models can be used for a hazard characterization of chemicals as well as for screening, the prioritization of chemicals and as supporting evidence for a risk-assessment approach.

Moreover, these models offer an innovative perspective for the risk-assessment tools developed by the authors and their colleagues, in the context of the LIFE VERMEER project (https://www.life-vermeer.eu/, accessed on 3 May 2022).

Finally, as we also previously highlighted within the text, some limitations were present in this work due to the intricacy of the experimental design, the dependency of the dose administered and the complexity of the toxicity mechanisms behind the adverse effects. It is worth underlining that, in the future, these models could be refined, applying the concept of the BMD, which is more reliable than NOAEL, since it is less influenced by the sample size and selection of the treatment dose. Moreover, with the awareness of their importance to refine the prediction of repeated-dose toxicity, a deeper evaluation of relevant issues, such as the role of the metabolism and mechanism of actions, will be taken into account in order to increase the work we are doing to provide advanced strategies for risk assessment and regulatory purposes.

## Figures and Tables

**Figure 1 ijms-23-06615-f001:**
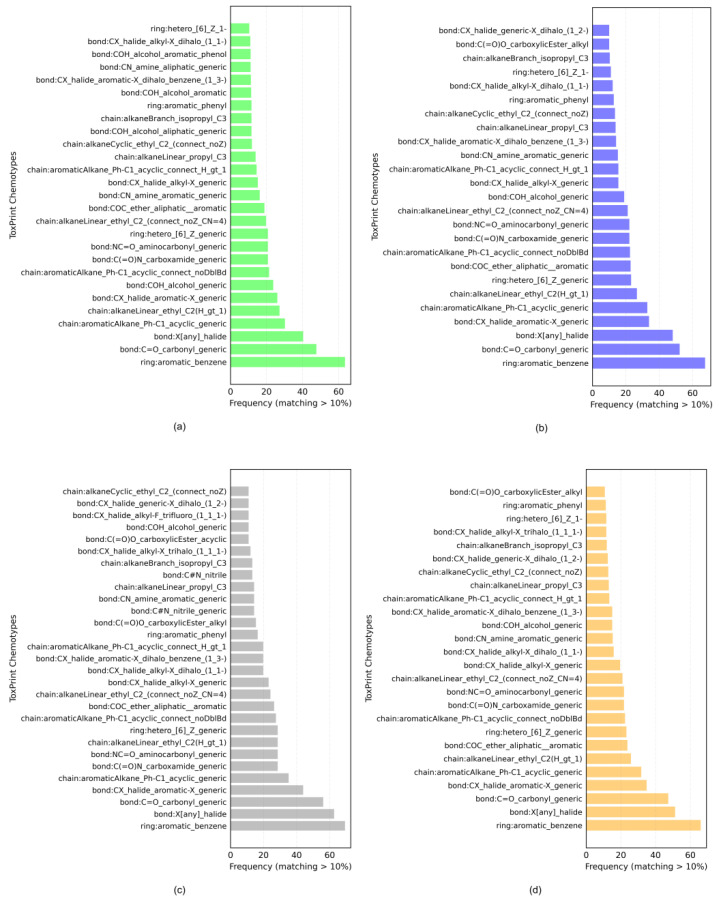
Histograms of ToxPrint chemotypes identified in the (**a**) general, (**b**) kidney, (**c**) brain and (**d**) liver datasets.

**Table 1 ijms-23-06615-t001:** The statistical characteristics of the models.

Model	Set	n	R^2^	CCC	IIC	Q^2^	Q^2^_F3_	RMSE	MAE	F
General										
NOAEL										
	active training	140	0.51	0.67	0.64	0.50		0.72	0.60	142
	passive training	140	0.51	0.69	0.65	0.50		0.81	0.69	142
	calibration	140	0.56	0.71	0.75	0.54	0.82	0.47	0.37	174
	validation	141	0.55	0.69	0.56		0.53	0.65	0.45	172
LOAEL										
	active training	142	0.55	0.71	0.70	0.54		0.66	0.50	173
	passive training	142	0.55	0.68	0.51	0.53		0.72	0.57	168
	calibration	137	0.51	0.63	0.71	0.49	0.48	0.71	0.56	138
	validation	137	0.53	0.63	0.59		0.32	0.72	0.53	149
Kidney										
NOAEL										
	active training	95	0.50	0.67	0.67	0.48		0.56	0.48	94
	passive training	95	0.52	0.54	0.57	0.51		0.73	0.64	103
	calibration	45	0.62	0.78	0.78	0.58	0.86	0.32	0.27	70
	validation	45	0.59	0.75	0.70		0.84	0.33	0.27	62
LOAEL										
	active training	97	0.51	0.68	0.62	0.49		0.51	0.40	99
	passive training	102	0.59	0.58	0.72	0.57		0.52	0.44	143
	calibration	46	0.67	0.81	0.82	0.63	0.77	0.35	0.28	88
	validation	38	0.69	0.82	0.66		0.81	0.31	0.25	80
Brain										
NOAEL										
	active training	23	0.55	0.71	0.57	0.46		0.67	0.59	26
	passive training	22	0.61	0.42	0.64	0.52		0.75	0.68	31
	calibration	23	0.58	0.74	0.76	0.51	0.88	0.31	0.26	28
	validation	22	0.53	0.67	0.68		0.81	0.35	0.28	23
LOAEL										
	active training	22	0.54	0.70	0.51	0.44		0.61	0.55	23
	passive training	23	0.69	0.42	0.83	0.61		0.91	0.82	48
	calibration	22	0.66	0.76	0.81	0.59	0.84	0.34	0.30	39
	validation	23	0.69	0.80	0.80		0.82	0.33	0.26	46
Liver										
NOAEL										
	active training	97	0.76	0.86	0.72	0.75		0.38	0.29	297
	passive training	94	0.73	0.83	0.78	0.72		0.44	0.34	247
	calibration	30	0.78	0.83	0.89	0.75	0.82	0.35	0.28	103
	validation	31	0.55	0.73	0.54		0.55	0.54	0.38	35
LOAEL										
	active training	96	0.78	0.87	0.78	0.77		0.32	0.23	326
	passive training	95	0.78	0.81	0.65	0.77		0.40	0.32	323
	calibration	30	0.76	0.84	0.87	0.72	0.71	0.38	0.28	89
	validation	31	0.61	0.71	0.61		0.55	0.48	0.40	46

**Table 2 ijms-23-06615-t002:** Comparison of QSAR models for NOAEL and LOAEL in rats for sub-chronic repeated-dose toxicity (90 days exposure) in the literature.

Model	Set	n	R^2^	Q^2^	CCC	Q^2^_F3_	RMSE	MAE	F	Reference
NOAEL										This work
	active training	140	0.51	0.50	0.67		0.72	0.60	142	
	passive training	140	0.51	0.50	0.69		0.81	0.69	142	
	calibration	140	0.56	0.54	0.71	0.82	0.47	0.37	174	
	validation	141	0.55		0.69	0.53	0.65	0.45	172	
	validation in AD (77%)	109	0.61		0.74	0.69	0.52	0.38	171	
LOAEL										This work
	active training	142	0.55	0.54	0.71		0.66	0.50	173	
	passive training	142	0.55	0.53	0.68		0.72	0.57	168	
	calibration	137	0.51	0.49	0.63	0.48	0.71	0.56	138	
	validation	137	0.53		0.63	0.32	0.72	0.53	149	
	validation in AD (74%)	102	0.51		0.64	0.55	0.58	0.45	105	
NOAEL										[6]
	training	124	0.57	0.56			0.68	0.52	164	
	invisible training	126	0.50	0.49			0.77	0.59	125	
	calibration	38	0.61	0.55			0.67	0.49	56	
	validation	38	0.65				0.68	0.52	68	
	validation in AD (87%)	33	0.69				0.58	0.43		
LOAEL										[6]
	training	124	0.55	0.53			0.66	0.51	147	
	invisible training	126	0.45	0.43			0.80	0.63	102	
	calibration	38	0.61	0.51			0.69	0.51	49	
	validation	38	0.59				0.72	0.54	53	
	validation in AD (87%)	33	0.62				0.59	0.44		
NOAEL										[10]
	training	97	0.53	0.51			0.61	0.47	107	
	Test	16	0.73	0.67			0.49	0.37	38	
	validation	27	0.60				0.43	0.36	38	
	validation in AD (96%)	26	0.61				0.42			

**Table 3 ijms-23-06615-t003:** Datasets examined.

ID	Model	Initial Number of Compounds	Final Number of Compounds	Final Range Value mg/kg bw/Day (log)
M1	General NOAEL	573	561	−1 to 3.591
M2	General LOAEL	573	558	−0.415 to 3.948
M3	Kidney NOAEL	289	280	0.097 to 4.301
M4	Kidney LOAEL	289	283	0.097 to 3.585
M5	Brain NOAEL	91	90	0.248 to 4.301
M6	Brain LOAEL	91	90	0.248 to 4.301
M7	Liver NOAEL	353	252	−0.432 to 3.499
M8	Liver LOAEL	353	252	−0.415 to 3.789

**Table 4 ijms-23-06615-t004:** SMILES and molecular graph attributes.

Attribute	Description
S, SS, SSS	Single SMILES atom, a combination of two SMILES atoms and a combination of three SMILES atoms [36,37].
EC1, EC2, EC3	Morgan connectivity of first-, second- and third-order [38].
NNC	Nearest neighbors codes [33,39].
APP	Atom pair proportions weighted presence of F, Cl, Br, N, O, S, P, #, = [40].

**Table 5 ijms-23-06615-t005:** Descriptors applied to build the models M1–M8 listed in Table 3
^(+)^.

ID	α	β	γ	δ	x1	x2	x3	y1	y2	y3	y4	IIC_w_	CII_w_	T*	N*
M1	1	0	1	0	1	1	1	1	1	1	1	0.25	0.30	1	30
M2	1	0	1	0	1	1	1	1	1	1	1	0.20	0	1	30
M3	1	0	1	0	1	1	1	1	1	0	0	0.25	0	1	10
M4	1	0	0	0	1	1	1	0	0	0	0	0.20	0	1	10
M5	1	0	0	1	1	0	0	0	0	0	0	0.50	0	1	33
M6	1	0	0	0	1	1	0	0	0	0	0	0.20	0	1	4
M7	1	1	0	0	1	1	1	0	0	0	0	0.15	0	2	15
M8	1	1	0	0	1	1	1	0	0	0	0	0.25	0	2	14

^(+)^ α, β, γ, δ are DCW indices; x1,…,y4 are equation constants; IIC_w_, CII_w_ are the correlation index weights; T* is the best threshold and N* is the best number of epochs.

## Data Availability

The data presented in this study are available in Appendix A.

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
