# Peer review of "Monte Carlo Models for Sub-Chronic Repeated-Dose Toxicity: Systemic and Organ-Specific Toxicity"

_ijms, 2022, doi:10.3390/ijms23126615_

Round 1
Reviewer 1 Report
The NOAEL and LOAEL prediction QSAR models on systemic and liver, kidney and brain were built using structure and toxicity data from public database in this work. This may be useful to assess the toxicity of the lead compounds in the process of drug discovery. It is a good try to build QSAR models to predicate sub-chronic toxicity. But, the significance of this work is not that much as author highlighted that “valuable alternative to animal testing”, because the predication data could not be used in the regulatory frameworks of REACH. It is suggested to revise the relative part before this work can be accepted for publication.
Reviewer 2 Report
The manuscript is quite interesting and challenging regarding the application field. I would have only a few general comments, which can be ou not be followed by the authors:
- as the models built up were based on molecular properties considering toxicity endpoints, not biological activity (related to drug-target interaction response), they should be called QSPR models, not QSAR;
- observing the Q2 and error values found, it seems that the models constructed for the liver organ data were the most consistent, and that result could be emphasized by the authors, pointing out important features related to the data set used.
Again, it is a very nice study, and in my opinion, could be expanded to heart and lung organs, as well.
